# Use of Mobile Phones in Classrooms and Digitalisation of Educational Centres in Barcelona

**Katia Pozos-Pérez [1], Gustavo Herrera-Urizar [1], Pablo Rivera-Vargas [1,2,*] and Cristina Alonso-Cano [1]**

1   Department of Teaching-Training and Educational Organization, Universitat de Barcelona, 08035 Barcelona, Spain
2   Faculty of Education and Social Sciences, Universidad Andres Bello, Santiago 8320000, Chile
*   Correspondence: pablorivera@ub.edu

**Abstract:** In the wake of the COVID-19 pandemic, multiple educational contexts experienced a sudden and accelerated digital transformation. However, this is not a new phenomenon. For years, public and private initiatives have been designed and tested in Spain. In this regard, the role and use of cell phones in the classroom has been a key and, at the same time, controversial aspect. In Barcelona (Catalonia), for example, recent educational policies have promoted the pedagogical use of cell phones. Within this framework, this article analyses whether these initiatives to promote the use of mobile phones are effectively transferred and implemented in the classroom. Using qualitative research, based on co-design, case studies and content analysis, we examined the reality of three educational centres in Barcelona. In these three contexts, field observations, interviews with management teams and ICT coordinators, and discussion groups with teachers were conducted. The information generated was grouped into five main categories of analysis. As a result, it was observed that the mobile phone has been losing prominence in the classroom. Schools tend to prohibit the use of cell phones and prefer computers to give priority to the control of technological tools in order to use the Internet safely. Mobile phones, in this sense, are only used at certain times when there is a pedagogical objective, although there is still a need for more pedagogical and digital training for teachers.

**Keywords:** mobile learning; mobile use; educational policies; educational practices; compulsory education; digitisation





## 1. Introduction

The evolution of mobile phones in the world has been explosive since the first call was made from the first mobile phone in 1973 [1]. Today, recent studies on mobile technology show that its use has been actively intensifying [2–4] and that this phenomenon will continue to grow. The main uses of these mobile devices are far from just making phone calls. Today, this evolution has meant that mobile phones can perform a huge number of simultaneous functions, boosted by the momentum and reach of the Internet, as well as the development of countless applications and the use of social networks. Thus, their main functions are now focused on socialising, conducting business, searching for information, shopping online, paying in stores, playing games, among many others, and of course, learning [5].

While this is not a new phenomenon, the truth is that with the COVID-19 pandemic the use of mobile devices has only grown, especially among the younger population. According to the recent report by Common Sense [3], the use of mobile phones by children aged 12 to 18 has increased by 17% since the pandemic began, and much more among adolescents (13 to 18 years). In the case of Spain, the most recent data state that Internet use is practically universal (99.7%) among people aged 16 to 24 and that 68.7% [4] of children aged 10 to 15 have a mobile phone. These data are very relevant and have intensified

the debate around the uses that children and young people make of this device, and the attitudes of families and the education system in general regarding its active use as a medium or tool for learning in schools [6,7].

Unlike other digital technologies, such as personal computers or laptops, which have been introduced and promoted as useful tools for learning and for the personal and professional development of students in the near future, mobile phones today pose a challenge when it comes to integrating them into the classroom [5,8]. While their ubiquity, their socialising function and their role in the development of digital skills are recognised, there is a clear fear that smartphones, due to their individualised and difficult to control usage, generate social inequalities and distractions that undermine the efforts of teachers [6,9].

UNESCO, for its part, advocates the appropriate use of mobiles in the classroom rather than banning them, arguing that in order to maximise the potential of information and communication technologies in education, we cannot ignore the mobile phone, a personal device that virtually all students have at hand, and proposes "to continue to harness mobile devices to support teachers and, by extension, improve learning opportunities for students around the world" [10] (p. 66).

In Spain, the lack of consensus on the issue at hand is also observed in the different political stances of the autonomous communities. Mellado-Moreno [11] refers to the existence of three different discourses. While the communities of Madrid, Castilla-La Mancha and Galicia have opted for prohibition, other autonomous communities have softened their positions, such as the Valencian Community and Aragon. Catalonia, on the other hand, through the mòbils.edu plan [12], is committed to promoting the use of mobile devices as a strategic educational tool for curriculum development, competence work, inclusive education, tutorial action and the management of coexistence and human relations to promote educational success [11].

Thus, the disparate pronouncements of the different autonomous communities contribute to increasing confusion within the education sector about how to deal with the fact that young people already routinely use this technology outside of school, beyond what they do in the classroom [13].

In this context, the project "Author" came about, whose main objective was to identify and analyse the discourses, practices and positions of educational administrations, teachers, young people, families and companies within the sector on the use of mobile phones in compulsory secondary schools in Spain. Based on an established classification prior to the implementation of the project, which placed the autonomous communities according to their political positioning on the use of mobile phones in the classroom (prohibition, promotion and indeterminacy), ten case studies [14] were conducted in compulsory secondary schools in four autonomous communities in Spain: four cases in Catalonia, which has promotion policies; two in the Valencian Community, which has indeterminate policies; and four in Madrid (2) and Castilla-La Mancha (2), which has prohibition policies. The purpose of the research has been to detect the forms of appropriation or reaction to the official discourses and to analyse the educational practices and dynamics that are promoted for the use of mobile phones in the classroom.

In the case of Catalonia, the fieldwork was carried out in three schools in the province of Barcelona and one in the province of Girona. This article presents the results of the three cases developed in secondary schools in the province of Barcelona (two public and one state-subsidised) that, in the first instance, were positioned as centres in favour of the use of mobile phones in the classroom and that had an explicit commitment to include mobile technology to promote learning processes and access to knowledge.

It is worth mentioning that the three schools analysed, in addition to what is established in their School Education Projects (SEP). The SEP is a document that includes the school's identity features, the pedagogical principles, the organisational principles and the linguistic project. This document specifies the values, objectives and priorities for action of the school, the curriculum and the cross-cutting treatment in the areas, subjects or modules of education in values and other teachings. In public schools, the SEP is drawn up by

the teaching staff, at the initiative of the head teacher. In state-subsidised schools, the head of the school approves the SEP, having listened to the school council (Departament d'Educació, Generalitat de Catalunya shorturl.at/acS38, accessed on 1 October 2022), have the Rules of Organisation and Functioning of the Centre (NOFC), which is a set of rules that regulates aspects related to: (1) the organisational structure and functioning of the school; (2) the participation of the school community in the life of the school; and (3) coexistence (the rights and duties of students, families and teachers). In this sense, the main research questions addressed in this article are: (1) Is there any promotion of the pedagogical or educational use of mobile phones in the classrooms of the schools analysed in the province of Barcelona? (2) Is there congruence between the schools' policies and regulations on the use of mobile phones (Discourses) and the practices carried out by teachers (Practices)?

Thus, in this article we present, after the analysis and interpretation of the research evidence, some answers to the questions raised from the reality within the schools themselves. All of this allows us to establish an overview of what happens at the different levels of concretion of the norms and the curriculum. It also allows us to reveal the different realities that are being shaped in educational practice from the analysis and interpretation of what is explicit in the SEP and in the NOFC, and discover what is really happening with regard to practices, contradictions, adaptations, successes, failures, fears and the potential of the use of mobile phones in secondary education in Catalonia as part of the digital transformation of education in the Spanish state.

## 2. Materials and Methods

The methodological framework addresses the need to respond comprehensively to the research questions outlined previously. This implies adopting a methodological perspective that allows us to understand and account for the transformations and implications of the digital society in the realities of schools. Thus, this article is the result of a qualitative research based on the development of three case studies (descriptive-interpretative) carried out in secondary schools in Barcelona, in which, according to [14], a contemporary phenomenon (the "case") is investigated in depth and within its real-world context, especially when the boundaries between the phenomenon and the context may not be clearly evident.

The criteria taken into account in the selection of the cases were: (1) public and state-subsidised compulsory secondary schools; (2) schools with an initial position in favour of promoting the use of mobile phones in education; (3) schools in which the educational use of mobile devices will be carried out; and (4) schools willing to participate in the study on a voluntary basis. Table 1 shows the profile of the participating schools:

**Table 1.** Profile of the educational centres of the study (case studies).

| | Centre 1—PV | Centre 2—BA | Centre 3—PF |
|---|---|---|---|
| Centre typology | Public institute | State-subsidised institute | Public institute |
| Mobile technologies utilized | Mainly Chromebook Laptops Smartphone (sporadically for educational use) | Laptops Smartphone (sporadically for educational use) | Chromebooks Smartphone (sporadically for educational use in the classroom or outside) |
| Main distinctive features of the educational centre | Compulsory Secondary Education and Baccalaureate. Educational project relates to the new methodologies used that imply responding to the "why" of 21st century education. | Pre-school, Primary and Compulsory Secondary Education. The fundamental axis of educational action is preferential attention to the needs of the student body. | Compulsory Secondary Education and Baccalaureate. It is committed to the participation of the entire educational community and is committed to the objective that its students achieve competency learning. |

In order to analyse the educational realities in depth, the research techniques were designed, and the instruments presented below were applied (Table 2).

**Table 2.** Research instruments and techniques by centre.

| Centre 1—PV | Centre 2—BA | Centre 3—PF |
| --- | --- | --- |
| 1 interview with the education management team/coordinator of the Educational Technology Department 1 discussion group with 3rd and 4th grade students 1 discussion group with teachers/tutors | 1 interview with the education management team 1 interview coordinator of the Educational Technology Department 1 discussion group with 3rd grade students 1 discussion group with teachers/tutors | 1 interview with the education management team and coordinator of the Educational Technology Department 1 discussion group with 3rd and 4th grade students 1 discussion group with teachers/tutors |

The process of designing the research instruments was based on collaborative work among the project participants. The starting point was the general research objectives and the specific objectives of each phase of its development. From there, the initial dimensions of analysis were defined and agreed upon by all members of the team, integrating the various contexts of implementation of policies and regulations (meso/institutional and micro/classroom). Subsequently, indicators were designed for each dimension to account for all the aspects to be investigated in the case studies, and these were specified in a matrix of dimensions and base indicators to elaborate the relevant items for each research instrument (Figure 1).

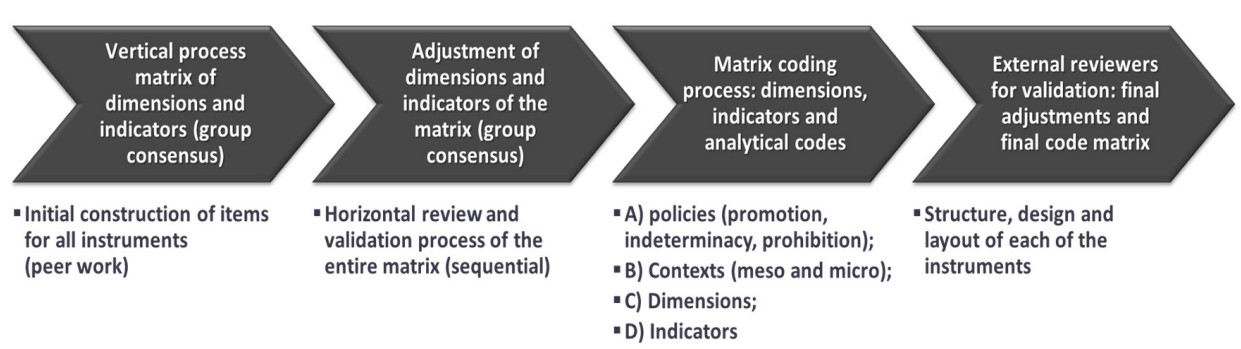

**Figure 1.** Research instruments and coding matrix design process.

The design of the instruments contemplated the integration of various sources of information, which allowed us to include the voices of the main educational agents in the case studies (Table 2) in order to subsequently carry out a triangulation of both sources of information and instruments and techniques for collecting information. In this sense, the items of each instrument were designed and adapted for each of the agents or sources of information: management team, teachers and students.

The data analysis was conducted by means of a content analysis understood as the set of techniques of analysis of the communications tending to obtain indicators (quantitative or not) by systematic and objective procedures of description of the content of the messages, allowing the inference of knowledge relative to the conditions of production/reception (social context) of these messages [15].

The units of analysis coded in the general analysis matrix (Table 3) were used as a starting point, and in which subsequently, emerging units of analysis arose from the in vivo coding, to identify and integrate the voices of the participants in the analysis. This in turn enriched the process and highlighted the way in which the intentions and actions of the people where the educational action was taking place were involved with the use of mobiles [16].

A transcendental aspect for the analysis was the need to structure it in such a way that it would shed light on the research questions posed, and for this reason, the analyses were grouped into: (a) Discourses, that is, what is clearly stated or made explicit in the policies and regulations of the centres; and (b) Practices, that is, what is really happening in educational practice for each of the sources of information consulted in the study.

**Table 3.** Final general code matrix for data analysis structure.

| Main Dimensions of Analysis | Centre 1—PV | Centre 2—BA | Centre 3—PF |
|---|---|---|---|
| 1. Regulations and policies/functioning of the Centre | | Discourses | |
| 2.Creation of materials/use protocols/Training | | | |
| 3. Freedom, democracy and educational co-responsibility | | Practices | |
| 4.Mobile phone uses, contexts and risks | | | |
| 5. Management of the centre and examples of good use of the mobile | | | |

## 3. Results

In accordance with the above, the results of this research are presented below. The first part (vision of the management teams of all the schools) includes the discourses and practices of the schools in relation to the regulations, protocols and management of the use of mobile phones. The second part (vision of the teaching staff of all the schools) includes the practices, the construction of materials and the appropriate use of the mobile phone in classroom contexts.

### 3.1. The Vision of Management Teams

3.1.1. Regulations and Policies of the Centre vs. Functioning

With regard to what is included and expressed in the regulations on the use of mobile phones in schools, it can be seen that all of them clearly state that they are in favour of the integration of digital technologies in educational processes and in the functioning of the school, and even present themselves as innovative schools that ensure the development of students' skills in accordance with the advances of the information society. However, not all of the schools have updated these rules, and one of them even argues that due to the pandemic, and all the issues that have had to be managed urgently because of it, the updating of the school's policies in general, not only on the use of mobile phones, has been deferred:

> *"Standards and policies have not been updated due to the pandemic. It is not currently considered a priority item in the management of the facility"* (C1).

In the School Educational Project (SEP), despite explicitly stating that it is in favour of the use of digital technologies, there is no specific mention of mobile phones.

In the specific operating regulations of each school (NOFC), it is clearly defined that the use of mobile phones in the classroom is not allowed without the consent of the teaching staff, and that students are penalised in different ways when they misuse them in any of the school contexts (classrooms, playgrounds, corridors or facilities). Only one of the three schools states that, in addition to keeping the regulations up to date, these are periodically and participatively reviewed and updated by the school's teaching staff, students and families; furthermore, this school explicitly states that it is in favour of the pedagogical use of mobile phones and any other digital technological device that contributes to learning:

> *"Proper and regulated use is encouraged in all spaces, and we have seen that incidences decrease dramatically and there is no need to penalise or remove mobiles [...] Measures are taken when there is evidence of misuse and with serious implications for students"* (C3).

This school has a clear and transparently disseminated policy for the entire educational community on the use of mobile phones at all times and in all contexts. It also encourages the use of mobile phones with families to manage the attendance of students or the participation of families in other school activities.

3.1.2. Creation of Materials and Protocols for the Use of Mobile Phones

In relation to the creation of training materials and actions focused on the use and pedagogical integration of digital technologies, especially mobiles, as well as protocols that

contribute to the good use of mobiles in the school and among the educational community, this is clearly a much less developed and systematised issue for most schools:

*"There are no policies for the transfer of this knowledge, material or training. There is training only in Moodle and a talk by the Mossos (Catalan police force) to families... Teachers in isolation share external resources that can help raise awareness"* (C1).

*"Training only on the platform of the centre: Google Suite and some of the Mossos (Police) to families"* (C2).

*"We have designed and adapted a colour protocol for the appropriate use of mobile phones according to the different contextual situations and we have posters and infographics in the corridors and in the different contexts of the centre [...] We do training in the centre, for families, for pupils, and we have a digital welcome plan"* (C3).

In general, the creation of didactic materials for the educational use of digital technologies in the classroom or the development of the digital competence of teachers and students is sporadic, and when it occurs, they are isolated experiences of teachers who develop these initiatives because they detect an educational need that is considered relevant.

### 3.1.3. Freedom, Democracy and Co-Responsibility in Education

The results of the analysis show very little freedom, in general, when it comes to promoting the democratic use of mobiles for educational purposes in schools. Their uses are very specific, and control and punishment predominate most of them. This translates into the preference of schools to use Chromebook or laptop computers in order to ensure strict control, even for the most inquisitive and innovative teachers, of the use of the Internet and programs or applications that can be used with a truly educational and motivational purpose. Adopting the Chromebook or laptop option is inherent in knowing the responsibility of families in the use of these devices:

*"As the family cannot control the use of leisure time on the mobile phone, they pass on to the school the responsibility to prohibit it, to act in a disciplinary way [...] Families say that they do not know what to do about the risks, such as bullying, and all that their children do with their mobile phones"* (C1).

Educational co-responsibility is not at all clear, since sometimes it is the families who transfer all the responsibility to the centre. Sometimes it is the students who delegate responsibility to the family and to the schools, and only some teachers and students recognise that it should be a shared responsibility.

### 3.1.4. Uses of Mobile Phones: Contexts and Risks

The management teams state that the use of mobile phones in schools is very sporadic, and they are only used for educational purposes when the teachers know how to do so. When they do not know how to use them, the possibility of investigating their educational and didactic potential is ruled out, so there is a great lack of knowledge of their potential in the teaching and learning processes, with some exceptions on the part of the teaching staff:

*"Students take pictures, make videos, upload everything to the Internet (to Youtube or playing games or making TikToks) in secret, in the corridors, the toilets, and in the playground"* (C1).

*"It is used in all contexts of the centre with colour regulation for reflective, positive, healthy and learning use"* (C3).

*"Exceptional use of mobiles is made according to the pedagogical needs of the teaching staff"* (C2).

In short, fear and a lack of knowledge predominate, especially in how to manage pupils' misuse and the risks involved. Along the same lines, school management teams state that the use of mobiles is very low with a few exceptions, but they are still far from exploiting their full educational potential:

*"It is rarely used only at specific times and by some teachers. Teachers are afraid and are largely unaware of the potential of mobile phones for education and learning"* (C1).

*"It is only used very sporadically by some teachers"* (C2).

*"Teachers recognise the potential of ICT and mobile phones in general. Their speed, immediacy, accessibility; they are the door to everything and everyone. They have a very high potential at the level of sensors and so they are suitable for the areas of science, technology and in physical education. They are also apt for detecting body parameters, location, and making calculations..."* (C3).

In addition, the mobile phone triggers conflicts that cause disruptive behaviour by students in schools. Even when the mobile phone is not used as a mediator of the teaching and learning processes, it can trigger risks of various kinds among students. Adolescents who find it difficult to manage themselves adequately, both within the school with teachers and students and outside the school with their families:

*"Children and adolescents are not prepared to manage a tool such as a mobile phone. They are not capable of managing this device in matters related to the violation of privacy"* (C1).

*"There are some uses and situations of mobile phone use that are not good. The compulsive use of social networks, addiction to games [...] but we recognise that it is not the problem of the mobile, but of the young person who has that addiction, something that is really worrying"* (C3).

*"Young people think that if they don't have a mobile phone, their parents are marginalising them"* (C1).

One of the key factors why mobile phones are possibly not being used in teaching and learning contexts in a generalized way is precisely the fear that each and every one of the educational agents has expressed; a distrust that prevents them from imagining a world of potential and possibilities. These suspicions show a lack of adequate and systematic training and display certain deficits associated with the limited digital competence of teachers as education professionals in the 21st century.

### 3.1.5. Management of the Centre and Examples of Proper Use of Mobile Phones

The results indicate that, without a doubt, the educational centres are concerned about the subject of mobiles and young people. They recognise that it is a technology that permeates the daily lives of all people but especially those of young people:

*"The centre is faced with the need to consider how to manage the use of mobile phones, as the families say that they cannot. Rather than banning them, which makes no sense, we have to teach the young people in their use and accompany them"* (C1).

*"We cannot ban it completely. It is clear that the mobile phone accompanies us in our daily lives. But let's see how we use it in a way that we do it well, both for school camps and for academic activities"* (C2).

In general, teachers believe that the use of mobile phones by students in schools must be managed in some way. However, they do not find a way to do this adequately, and therefore, in the face of fear, management is oriented towards control and penalisation. Although some schools state that they do need adequate training and knowledge to manage the need to improve their relationship with the use of mobile phones as a school, they also recognise the need to organise training with their teaching staff and students, and also with and for their own families.

Even though schools in general are reluctant to use mobile phones in the classroom, we have mentioned that there is evidence that some teachers in various subject areas are exploring the educational uses of mobile phones and some applications that are really beginning to enhance student learning:

*"In the humanistic itinerary we made a practical trip with Maps to visit the spaces of the Civil War in Barcelona. Every 2–3 students prepared a SPAR Route in which they had*

*to take a picture of the place and explain to their classmates what this space was. Thus, visiting and explaining historical places in which each place was geolocated, having an itinerary with Maps, uploading a photo, making a summary, sharing with the tutor of the subject, and giving credit to the people with whom you had collaborated, is an example and an indisputable guarantee of what is a good didactic use with the mobile"* (C1).

*"We created WhatsApp groups for the management and coordination of the centre, as well as an APP for attendance control and communication with families, among other functions. We also implemented a Clickedu type management platform for the centre"* (C3).

Finally, it should be pointed out that there is the professional development of the teachers themselves, which is positively valued by the school management, and it is considered that they should always be at the forefront in the supervision of the students.

### 3.2. The Vision of the Teaching Staff

### 3.2.1. Regulations and Policies of the Centre/Functioning

The teaching staff of the participating schools stated that they are aware of the school's regulations on the use of digital technologies in a broad sense, and of mobile phones in particular. However, some teachers say that there are contradictions between what is recommended at the level of the Generalitat de Catalunya, the practices of the centre, and what actually happens in each of the classrooms in each specific area of knowledge. They consider that teachers should adapt the regulations at all times depending on the group of pupils, their characteristics and their particularities:

*"We know the rules of the centre, and we agree, but it is very difficult to carry them out or to apply them. We try by all means. I think that all these measures that every teacher takes in our groups are for a reason. I think that the use of mobile phones and computers is good, it's a tool, but they misuse the computer and the mobile phone, and that's why we take these measures. We all follow the internal rules of the school, but when the mobile phone is something personal and the rules say that you can't touch it, then you can't do anything with their computer or mobile phone"* (C1).

*"There is a contradiction between the regulations established by the Department of Education and the school's regulations and what can actually be done in the classroom with mobiles and technology. It is a very restrictive vision in which the department itself organises training for the educational use of some social networks or mobile phones and then blocks it. Therefore, teachers do not have "so much freedom" to be creative or innovative with ICT or mobile phones. They give the option of being able to activate or deactivate the permission to use "minijuegos.com" but not TikTok, for example, which can be used for educational purposes, and from which teachers learn to reflect critically on its use with their students"* (C3).

Some teachers are even more restrictive than the school to avoid disruptive behaviour, and others, occasionally or sporadically, are the ones who explore the practice of using mobile phones for the benefit of their students' learning.

### 3.2.2. Creation of Materials and Protocols for the Use of Mobile Phones

The creation of didactic materials for the educational use of mobile phones or the drafting of protocols for action at the school regarding the proper use of mobile phones is practically non-existent. Only in one of the schools (C3) are materials and protocols for the use of mobile phones created and disseminated within the school (classrooms, playground, corridors) and among families through information sessions at the beginning of the school year:

*"They know the school rules. In each classroom there is an explanatory sign with four colours: (1) red, which indicates that you cannot use the mobile phone because the teacher is explaining or does not give permission at that moment and you cannot*

*use it; (2) yellow, which means that you can use the mobile phone if the teacher gives permission; (3) blue, which means that you can only use it to look for information in classrooms, laboratories and workshops with the teacher's permission; and (4) green when participating in activities organised outside the centre, as long as it does not interfere with teaching activities"* (C3).

In practically all of the schools, teachers say that to a greater or lesser extent they need training, not only in relation to the use of mobile phones in education but also, in a broad sense, in digital teaching skills: training that allows them to make use of the diversity of digital processes for managing their own teaching. Some centres conduct specific training on the use of their educational platforms, such as Moodle, but it is not a continuous or systematic training:

*"I would say that each one of us, individually, has been able to be trained, but I don't think we received training in ICT"* (C1).

*"In July we attended a rather boring training on the uses of mobile devices in the classroom. I think that what we need is competence training. What we received was a classic training, which I find very incoherent and very impractical. In the school we have a regulation that states which social networks we must limit, but we attended a training session in which we were encouraged to use social networks. It was a waste of time. We are acquiring digital competence little by little. We share our experiences with each other, which I consider very positive"* (C3).

The teaching staff also say that on sporadic occasions, and at the initiative of a teacher, they organise small training sessions, but this is not an institutionalised practice. The same happens with the creation of educational materials related to the use of digital technologies or the good use of mobile phones. Very few teachers create and share resources with their colleagues.

### 3.2.3. Freedom, Democracy and Educational Co-Responsibility

There is much fear of the addiction and misuse of mobile phones by students. Educational centres attempt to control the situation and manage or negotiate the conflict by trying to take away the students' mobile phones. In that sense, each teacher adjusts the rules, depending on the type of group and their behaviour:

*"I think that the mobile phone regulations, properly adjusted and understood, are very much in line with our way of doing things. We give teachers a lot of freedom to use this device if they think it is convenient, or if an activity requires it, but it is true that this situation has been perverted a bit. This freedom is sometimes misunderstood, and I think that we have crossed a limit and that the use of the device is being misused a little. When you enter the classroom, this device should not be present. It should only be present when the teacher requires it. That way we would avoid sanctions, discomfort and annoyance, both among teachers and students, because sanctions always lead to discomfort, both for the teacher who has applied them and for the students who receive them"* (C3).

In general, teachers consider that students do not have the capacity or the responsibility to make good use of mobile phones, and they consider that families do not know how to do so either and pass the responsibility on to the schools.

### 3.2.4. Uses of Mobile Phones: Contexts and Risks

It is confirmed that the mobile phone is rarely used for educational purposes and only at very specific times, both in the classroom and on school outings, but always under the supervision of teachers. In the playground, mobile phones are prohibited in some schools, and in others they are closely supervised. Even though personal use within the school premises is forbidden in all the schools, it is observed that students always find a way to use the device in secret from teachers and not for educational purposes:

*"They use it before entering the school, before starting classes, some in the corridors on the sly. In the playground, it is allowed but with school supervision, and in the classroom, they always have it at hand, but it should only be used when indicated by the teachers to carry out a class activity. At a pedagogical level, I have seen very interesting activities and dynamics that have been done with the mobile phone. I personally use it in physical education classes"* (C3).

In the cases in which it is used, either in the classroom or on outings, it is confirmed that there are several disciplines in which the potential of the device is explored, such as science, physical education, language and in the reception classroom; however, the great lack of knowledge is reiterated, as is the need for training on the potential of the mobile phone in education.

We can also say that the teachers of all the schools participating in the study agree on the enormous amount of risk associated with the misuse of mobile phones and the need for training of teachers, students and families in their good use, not only educationally but also ethically:

*"Students spend many hours on the screens, and this can affect their health. The hours they use screens at school, plus the hours they use them at home or in their daily lives, are many. They have an obsession with looking at a screen to feel safe. There is a very high dependence on mobile phones, and so they always need to have them in their hands"* (C1).

*"We have been working all these months on developing a regulation on cyberbullying. We are concerned about the obsession with showing off on the networks, and exhibition-ism"* (C2).

*"If you leave them alone, they're on social media, WhatsApp, video games, looking at their Instagram account, or personal stuff. They don't make good use of it. In the hallways they sneak a peek. In the playground they isolate themselves quite a bit by playing video games. Even when using it in class for educational purposes, they tend to use it when it's not their turn or for other things. They can't live without their cell phone. Every time you have tried to take it away from them, there are some students who get very angry and don't want to give it up"* (C3).

It is evident that there is much fear of all the risks related to the use of mobile phones. Most teachers have had among their students cases of addiction, excessive use, cyberbullying and identity theft, among many other conflicts that they do not know how to deal with.

### 3.2.5. Harnessing the Potential of Mobile for Learning and Good Mobile Use

Linked to the previous aspect and given the low use of mobile phones in the classroom, it is difficult to speak of a real use of their potential in the teaching and learning processes. Some teachers recognise that increasing the use of mobile phones in class could be something positive and necessary, while others prefer to follow traditional teaching methodologies. Some consider that it is not necessary to use the mobile phone in class proposals and activities because laptops or Chromebooks are already being used, while others argue that the ideal would be to be able to use all available devices correctly:

*"I think it would be ideal to use the mobile phone and the computer and all the tools that make our work easier and with which we can do research, but the problem is that the students don't make good use of the mobile phone or the computer because they are not responsible. Ideally, everyone should have a computer and a mobile phone, and everyone should use them for what they are supposed to use them for. But they arrive with iPods and headphones, and they don't listen to you. Many times, no matter how much you want to innovate, using platforms or mobile phones, you reach a point where you realise that the only way to see if they have learned or not is to go back to the old ways, and then you wonder what's the point of investing in new technologies since they, deep down,*

> *know more than me. It is a generation that was born with these technologies, and we are learning as we go along"* (C1).

Although the mobile phone is used very occasionally for educational purposes, there are some very interesting and beneficial pedagogical experiences for the development of skills among students and teachers:

> *"The mobile phone is very necessary in the reception classroom as it allows them to use the simultaneous translator if they need it. It is a tool that allows them to reinforce their autonomy inside and outside the school, so I think it is very important that they learn to use the mobile phone"* (C1).

> *"We use the mobile phone in physical education to record and self-evaluate. In social sciences we use it for virtual reality practices using Google Cardboard and Oculus. We also use apps to measure air pollutants, and Arduino Science Journal, to measure decibels and assess noise pollution"* (C3).

> *"The mobile phone is very useful in the case of students with special educational needs. It is a tool which, if they know how to use it, helps them to take a step forward in accessing information. It allows them to translate, or listen, if they cannot read"* (C1).

These experiences of using the mobile phone, although very positive, are unique and there are no established spaces in which to share and democratise them.

## 4. Discussion

The discussion has been organised based on the most relevant issues arising from the analysis. Here a dialogue will be established between the positions of the management teams and teachers, and the arguments derived from the theoretical framework of the project.

### 4.1. The Management Team and Mobile Phones

The management teams of the three schools are reluctant to use mobile phones in the classroom. This explains to a large extent why the regulations and policies of the schools focus on prohibiting the use of mobile phones. There is a great lack of knowledge of the potential of these devices as mediators of the teaching and learning processes.

The main fear factor observed is the lack of knowledge of risk management involving the inappropriate use by young people, who also do not receive training and guidance. This leads to fears of potential conflicts among students [17]. The management also perceives the pressure from families who seem to shift the responsibility for this issue to the school, also due to a lack of knowledge of how to deal with it with their children [18].

### 4.2. Teachers and Mobile Phones

Teachers, in general, also claim to have certain fears and insecurities when it comes to facing and managing the uncontrolled use of mobile phones by students. They repeatedly state that with each passing day the risks are higher, more frequent and have greater reach, and the speed at which the damage spreads is increasingly faster. All of this combines with a lack of techno-pedagogical training to take advantage of the educational possibilities of mobile phones [19]. Faced with this situation, in most cases, teachers opt for excessive surveillance and control of everything students do with their devices at all times. There are teachers who, in an isolated and unique way, dare to explore the potential and possibilities of mobile phones in the classroom: teachers who are conducting powerful learning experiences, even in schools with the most restrictive regulations. The considerable potential to improve and facilitate learning with mobile technologies is widely recognised, among others, by UNESCO (2017); however, it is not something obvious and easy to achieve and there are very few teachers who carry out pedagogical practices with mobile phones that enhance the learning experiences of students. Practices that, according to research evidence, work well, increase motivation for learning and content, and foster collaborative work, creativity and coexistence. Within the recommendations of

UNESCO [10], there is an explicit understanding that educational interventions in the use of mobiles should be integrated into carefully planned projects that should go far beyond the technology itself. Therefore, the appropriate use of mobiles in the classroom and the development of appropriate policies for it should take into account timely pedagogical training (development of digital teacher competence), both for teachers in training and active teachers throughout their professional development, as well as for students and families, in response to the challenges of each particular educational context. We agree with UNESCO [10] that the appropriate use of mobiles in the classroom complements and enriches formal and non-formal education, insofar as it contributes to making learning more accessible, equitable, personalised and flexible for learners around the world.

Fombona and Rodil state that:

*"Mobile devices have a reduced use as a teaching and learning tool in classrooms, although most teachers and students would like to use it more often. Possibly there are still some difficulties, fears and methodological ignorance regarding its use, to be able to implement it as a common working tool in the classroom. Its growth and evolution have been so rapid that perhaps we have not yet had enough time to take advantage of its potential"* [6] (pág. 32).

### 4.3. Laptops vs. Mobiles

The evidence of this research has allowed us to observe that the mobile phone has gradually been losing prominence in the classroom. In this sense, it is noted that both teachers and students tend to prefer the use of Chromebooks and laptops in order to prioritise the control of resources and safe Internet browsing. Mobile phones, therefore, represent a pedagogical and instrumental challenge, more than an opportunity, when integrating them into the classroom [5,8]. Thus, while it is impossible to eliminate the use of mobile phones, since it is part of the daily life of students and teachers, there is a clear need to generate protocols in the educational community that aim to reduce their potential as a distracting element and favour their sensible and constructive use.

### 4.4. The Market vs. the Educational Curriculum

The market stimulates the use of technological resources that modify and/or transform the implementation of the educational curriculum [6]. For example, in some centres they do not use mobile phones, but they use similar tools such as educational platforms or Virtual Learning Environments on portable devices. In centres 1 and 3, they mainly use Chromebook, laptops and mobiles very sporadically for pedagogical use. In centre 2, they use mainly laptops and again mobiles sporadically in the classroom. Moodle is used as the preferred platform for subject management in all the centres, but its use is also combined with the use of Google Suite, especially for centre and teacher management, as well as communication between the teaching staff. Thus, the market offers different resources that schools have acquired without the necessary understanding of all the implications that this type of decision entails.

### 4.5. Fear and Privacy Risks

In our fieldwork, we have observed that there is a clear lack of knowledge on the part of the educational community about the uses that students make of mobile phones outside of schools. The truth is that families hand over a mobile device to increasingly younger ages [3]; however, neither they nor the children and adolescents have the knowledge and skills that allow them to make safe use of the device and understand the risks to which their privacy is exposed. In view of this, it is essential that families, students and the school community as a whole agree on a sensible, responsible and reasonable use of mobile phones both inside and outside school.

## 5. Conclusions

The conclusions have been organised in order to answer the two main questions posed in the introduction and which have guided the development of this article.

In relation to the first question: Is there a promotion of the pedagogical or educational use of mobile phones in the classrooms of the schools analysed in the province of Barcelona? We can affirm that even though the schools were initially selected based on their explicit stance in favour of the use of technology and mobile phones, which can be found in their regulations, we can conclude that there is no clear attitude to promote the use of these devices, largely due to the uncertainties generated by their management with the pupils, and because it is considered to be a distracting element in the classroom. In fact, their use is limited to certain sporadic pedagogical actions, which are not systematic and are not reflected in the curriculum or the teachers' continuous educational planning.

It is also observed that the mobile phone has been losing prominence in the classroom. Schools tend to prohibit the use of mobile phones and prefer computers to work with digital platforms and media.

In relation to the second question: Is there congruence between the policies and regulations of the schools in the use of mobile phones (Discourses) and the practices carried out by teachers in educational practice (Practices)? In the development of this research, we have been able to appreciate that there is a gap between (1) what is established in the autonomous regulations of Catalonia and in the guidelines of the schools themselves, and (2) what happens concretely in teaching practice. From our initial hypothesis, which stated that there was a promotion of the use of mobile phones in schools supported by regional policies and their regulation, we see that, in reality, what generally exists is a ban on the use of mobile phones. On the other hand, their use is only promoted when a pedagogical objective is defined by the teaching staff, which is something unusual in the development of teaching practices.

We would like to emphasise that the conclusions reached in this research must also consider their limitations. In this respect, one of the main limitations has to do with the representativeness of the centres, that is to say, that the selected centres can really express all the educational diversity that exists in Barcelona, as well as other centres that integrate the different territories of the city itself. In addition, as a result of the COVID-19 pandemic, and the consequent work overload experienced by the educational centres from 2020 onwards, we had substantial complications in accessing them. Beyond this, the purpose of this research and consequent article has not been to close the analytical focus on the subject, but rather to help broaden the understanding of the phenomenon and encourage other research exercises along this line.

**Author Contributions:** Conceptualization, P.R.-V.; methodology, K.P.-P. and G.H.-U.; validation, P.R.-V., K.P.-P. and G.H.-U.; investigation, K.P.-P., G.H.-U., P.R.-V and C.A.-C.; writing—original draft preparation, K.P.-P., G.H.-U. and P.R.-V.; writing—review and editing, P.R.-V. and C.A.-C.; funding acquisition, C.A.-C. All authors have read and agreed to the published version of the manuscript.

**Funding:** Grant "Young people and mobile phones in the classroom. Discourses and dynamics of prohibition, promotion, and indeterminacy" [PID2019-108041RB-I00] funded by MCIN/AEI/10.13039/501100011033.

**Institutional Review Board Statement:** Not applicable.

**Informed Consent Statement:** All participants in the study were volunteers and informed consent was given prior to the questionnaires.

**Data Availability Statement:** Not applicable.

**Acknowledgments:** We thank ESBRINA Research group, Contemporary Subjectivities, Visualities, and Learning Environments (2017SGR 1248) (http://esbrina.eu, accessed on 23 December 2022). We also thank REUNI + D University Network for Educational Research and Innovation, Connecting Networks and Promoting Open Knowledge, University Network for Educational Research and Innovation, Connecting Networks and Promoting Open Knowledge, RED2018-0180-102439-T (http://reunid.eu, accessed on 23 December 2022).

**Conflicts of Interest:** The authors declare that the research was conducted in the absence of any commercial or financial relationships that could be construed as a potential conflict of interest.

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
