# Peer review of "Use of Mobile Phones in Classrooms and Digitalisation of Educational Centres in Barcelona"

_education, doi:10.3390/educsci13010021_

Round 1

Reviewer 1 Report

The issue is relevant for the field and the manuscript is clear. It is valued that the manuscript is well-structured. It could be important highlighting the hypothesis at the beginning of the paper.

Author Response

Thank you very much for your positive feedback and advice, which we will take into account for future research in the same field of study.

Reviewer 2 Report

he article shows the results obtained in the execution of a research project carried out in the city of Barcelona in which the educational impact of the use of mobile phones in classrooms is analysed. The description of the context, together with the review of previous literature on the subject is relevant and adequate. The methodological design is well formulated, and its implementation, as described, is in line with the traditional parameters of educational research.

The results are relevant, as they show (against all odds) that the pedagogical use of mobile phones is losing relevance and importance. This is significant, especially in a city like Barcelona, characterised by being an active space for innovation and digital transformation in the classroom (where the Mobile World Conference is held every year), and a promoter of "BYOD" educational policies a few years ago. 

It would be very interesting to measure the extent of the phenomenon (as presented in the results) in other territorial contexts. It would also be useful to triangulate research techniques, for example by designing and implementing a questionnaire for teachers and students. But these may be challenges to be addressed in other research. What is presented in this article is enough to bring us closer to a problem that should be further explored by educational and digital policy makers and academia.

Author Response

(The authors gave the same response as above.)
